# Pseudosaccades: A simple ensemble scheme for improving classification performance of deep nets

## Abstract

We describe a simple ensemble approach that, unlike conventional ensembles, uses multiple random data sketches ('pseudosaccades') rather than multiple classifiers to improve classification performance. Using this simple, but novel, approach we obtain statistically significant improvements in classification performance on AlexNet, GoogLeNet, ResNet-50 and ResNet-152 baselines on Imagenet data – e.g. of the order of 0.3% to 0.6% in Top-1 accuracy and similar improvements in Top-$k$ accuracy – essentially nearly for free.

## 1 Introduction

Deep Neural Networks (DNN) are state-of-the-art tools for various machine learning tasks (LeCun et al., 2015) and have proved especially useful for image classification tasks. For example, the most recent winners of the Imagenet challenge [1] have all been DNNs.

Although it is not at all well understood – at least in terms of formal learning-theoretic guarantees – how and why DNNs perform so well [2], empirical understanding of how to construct a DNN is substantial and growing, and there are many plausible hypotheses regarding their performance. One striking example of the latter is that not only do DNNs have clear parallels with aspects of human visual processing, but in controlled psychological experiments they also match human performance on visual recognition tasks very closely (Serre et al., 2007).

Taking inspiration from nature, in this paper we show that an approximate analogue for saccades in human visual processing can improve the performance of a carefully-tuned DNN on an image classification task *that it was explicitly designed to solve*. More precisely, we use a very simple ensemble-like approach that employs voting but, unlike typical ensemble approaches, rather than learning several similar DNNs and obtaining a weighted combination of votes from that ensemble, instead we use just a *single* DNN but feed it as input multiple random low-dimensional sketches of an image and take the DNN's vote *with itself* on these sketches to reach a majority verdict.

Using our simple approach we obtain statistically significant improvements in classification performance on AlexNet, GoogLeNet, ResNet-50 and ResNet-152 baselines on Imagenet data – e.g. of the order of 0.3% to 0.6% in Top-1 accuracy – essentially nearly for free. We carry out a comprehensive empirical exploration of our approach, reporting results using different levels of subsampling and different ensemble sizes, as well as an initial exploration of whether the improvements have any identifiable systematic component (such as occurring disproportionately in the same class).

## 2 Motivation

Our approach is inspired by considering saccades in human visual processing, that is eye movements that focus attention on elements in a visual scene. The human eye has only a few degrees of visual arc

---

[1] http://www.image-net.org/challenges/LSVRC/

[2] For example, while it is known that the VC dimension of a DNN is upper-bounded by the number of nodes in the network (Anthony & Bartlett, 2009), it is not known why DNNs dramatically outperform 'wide' neural networks with the same number of nodes but fewer hidden layers.

of high-resolution imaging capability, and saccades are a mechanism by which a scene can be estimated from high-resolution subsampling of parts of it. In human visual processing this subsampling is not uniformly at random – we attend to certain features proportionately more often than others – but we hypothesised that an evolutionary precursor to saccades could have been something closer to a uniform random sampling of features in a scene and that (if indeed there was such a precursor) this must have conferred some selective advantage in order to propagate. Meanwhile recent theoretical results in Lim & Durrant (2017) show that randomly subsampling rows and columns from an image without replacement results in – with high probability – an approximately affine transformation of the original image. Putting these ideas together, since image labels should remain invariant under affine transformations, we speculated that such subsampling could potentially lead to improved classification performance, perhaps even for an already highly-accurate classifier, by providing the classifier with multiple low-dimensional sketches of the same image in a similar way that saccadic sampling of a scene does. We call this subsampling of rows and columns 'pseudosaccades'. Because image classification inputs can be of varying sizes, while most classification algorithms accept only a fixed size input, a common preprocessing is to convert them to a (usually smaller) standard-sized input prior to classification. However as far as we are aware it has not been much exploited before that such preprocessing offers an opportunity for generating multiple instances of a particular image. By extracting pseudosaccade sketches of an image before applying the standardizing preprocessing, allows the generation – for typical image sizes – of thousands of such instances *per image*. Moreover unlike cropping and reflection the resulting pseudosaccade images resemble photographs captured following a change of camera angle and position, while still keeping the subject central in scene – see figure 1. Confirming our conjectures, we found that exploiting several of these pseudosaccade sketches can, indeed, improve classification performance and in the remainder of this paper we present and discuss our experimental results.

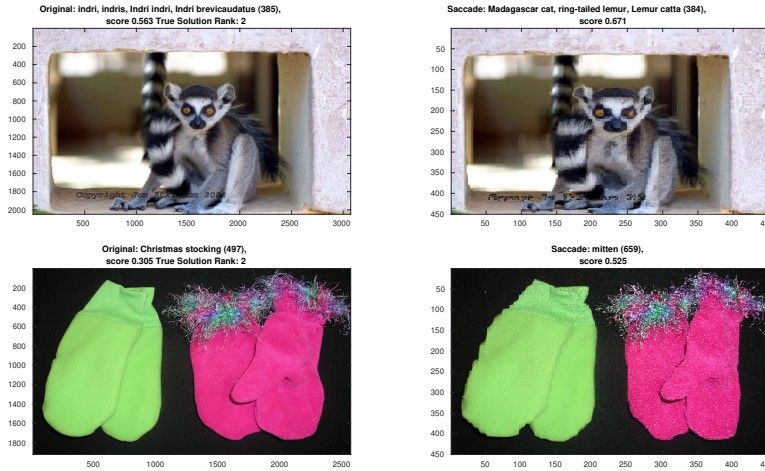

Figure 1: Two images incorrectly classified by Alexnet in their original form (left-hand column), but correctly classified in pseudosaccade form (right-hand column).

## 3 EXPERIMENTS AND RESULTS

In this section we present details of our experimental protocol and the results of our experiments. We show that the classification accuracy on a single pseudosaccade version of an image is similar to the accuracy on the original images, given a suitably high projection dimension. Moreover, using pseudosaccades as a diversity generator, an ensemble classifier employing several pseudosaccade versions of each image can consistently outperform the classification accuracy of the same classifier on the original images.

## 3.1 DATASET

We used the validation dataset from the Imagenet Large Scale Visual Recognition (ILSVR) Challenge 2012 as described in Russakovsky et al. (2015) for our experiments. This dataset comprises of 50000 images, ranging in size from 56x54 pixels to 5005x3646 pixels, where each image is an example from one of 1000 distinct classes. The subject of an image (i.e. its class label) is the dominant, and usually the central, object in that image and therefore elements of attentive viewing are already present in these images by virtue of the location of the subject. The classes in this dataset range from broad categories to fine-grained labels – for example one subset of the labels is a classification of 120 different breeds of dogs. Table 1 summarizes the main characteristics of this dataset.

## 3.2 CLASSIFIERS

We used the winners of ILSVR Challenge from 2012, 2014 and 2015 namely Alexnet (Krizhevsky et al., 2012), GoogLeNet (Szegedy et al., 2015), and ResNet-50 and ResNet-152 (He et al., 2016) to represent the state of art in deep neural network classifiers; These classifiers include many of the latest developments in the evolution of neural networks and each introduced new architectures and other innovations such as ReLU activation functions and skip connections, resulting in the highest accuracies on the Imagenet Large Scale Visual Recognition Challenge for the years 2012, 2014 and 2015 respectively. We used the MATLAB versions of these DNNs implemented in Matconvnet (Vedaldi & Lenc, 2015) and we used the pretrained weights which are tailored for the ILSVR task to provide a consistent baseline. We note that the pretrained weights for GoogLeNet uses weights from Princeton instead of Google, which may affect the accuracy for this DNN compared to the challenge-winning DNN. Also published accuracies in Krizhevsky et al. (2012); Szegedy et al. (2015); He et al. (2016) for the ILSVR challenge are on the challenge test dataset, while we used the validation dataset because it has the labels available. Thus our accuracies for these DNNs show some discrepancies with those published results. Table 2 is based on a similar table from Alom et al. (2018) and summarizes the characteristics of these DNNs as well as the baseline accuracies we obtained on the ILSVR challenge validation dataset using them.

## 3.3 EXPERIMENTAL PROCEDURE

To obtain baseline accuracies for each of the four DNNs we classified each image in the ILSVR validation set with no preprocessing, other than that implemented by the DNN itself to standardize the image sizes. The preprocessing carried out by the DNNs is noted in table 2. We measured the top-1, top-3 and top-5 accuracy for each classifier on the full validation set of 50000 images – these are also presented in table 2 and we will refer to these results obtained on the original images (without subsampling) as the 'baseline classifier' results.

For our pseudosaccade approach we first fix the 'projection dimension' to be an integer $d \in \{450, 430, 410, 390, 370, 350, 330, 310, 290, 270, 250, 200, 150\}$ and then randomly sample $\min(d, width)$ columns and $\min(d, height)$ rows from the images without replacement. As in the baseline experiments, we apply no further preprocessing, other than that implemented by the DNN to standardize input size, and we measure the top-1, top-3 and top-5 accuracy for each DNN on all 50000 images in the ILSVR validation dataset. We refer to these results as the 'saccade classifier' results. We also store the scores and the top-5 predicted labels for each combination of sampled projection dimension $d$, image, and DNN. Since the obtained accuracies, scores, and labels are realizations of random variables we repeated these experiments for each combination of $d$, image, and DNN a total of twenty-four times, and we calculated the means and standard deviations for the top-1, top-3 and top-5 accuracies.

Keeping $d$ fixed we construct an ensemble of size $m \in \{1, 2, \ldots, 15\}$ using the scores of between one and fifteen saccade classifiers by sampling without replacement $m$ sets of top-5 scores from the 24 sets of stored saccade classifier scores. We combine these to obtain the ensemble decision by simply summing scores for each label. For each $d, m, k$ triple and each classifier we repeated this process fifty times and we calculated the corresponding means and standard deviations for the top-1, top-3, and top-5 accuracy.

Table 1: Properties of the Imagenet validation dataset

|  | Min | Mean | Max |
| --- | --- | --- | --- |
| Image Count | | 50000 | |
| Label Count | | 1000 | |
| Fine-Grained Labels | | 120 | |
| Height | 56 | 430.25 | 5005 |
| Width | 54 | 490.37 | 4288 |
| Size | 3456 | 231320 | 18248230 |

Table 2: Summary of the DNN classifiers

|  | AlexNet | GoogLeNet | ResNet-50 | ResNet-152 |
| --- | --- | --- | --- | --- |
| Architecture | CNN | LeNet-Inception | Residual Neural Network | |
| # Convolution Layers | 5 | 57 | 50 | 152 |
| # Fully Connected Layers | 3 | 7 | 1 | 1 |
| # Parameters | 61 M | 7M | 25.6M | 60.3M |
| # Multiply and Accumulates | 724M | 1.43G | 3.9G | 11.3G |
| Regularization | Batch Normalization | Local Response Normalization | Batch Normalization | |
| Image Resizing | bicubic scaling (227x227) | bilinear scaling (224x224) | | |
| Top-1 accuracy | 54.70% | 65.46% | 70.39% | 72.45% |
| Top-3 accuracy | 71.68% | 82.22% | 85.55% | 87.05% |
| Top-5 accuracy | 77.56% | 86.93% | 89.66% | 90.66% |

## 3.4 RESULTS

Baseline results for the four DNNs are given in table 2. Results for our pseudosaccade classification ensembles are given in tables 3 and 4 for ensembles of size 5 and 10 respectively and as well they are plotted in figure 2 for all values of $d, m$ and $k$. In figure 2 the orange plane shows the baseline accuracy for each classifier and top-$k$ combination within a sub-figure. The surface plots show the average classification error for a given $d, m, k$ triple using pseudosaccades. From tables 3 and 4 we see that these average outcomes are very stable indeed, and if the projection dimension $d$ is sufficiently high then even a small ensemble can outperform the DNNs working with the original images at the 5% level of significance (or better) on Top-1, Top-3 and Top-5 classification accuracy. On the other hand we see that using a single pseudosaccade representation of each of the images, although we can match or nearly match the baseline accuracy with a projection dimension as high as $d = 350$ (see figure 2), with a lower projection dimension we obtain far worse accuracy than the baseline. The curve comprising the left-hand boundary of each surface plot shows the average accuracy for a single pseudosaccade plotted against the projection dimension $d$. Finally we see that the accuracy of the ensemble exceeds that of the baseline classifiers, even for a small ensemble of classifiers and small projection dimension, and this behaviour is consistent across all of the classifier architectures.

## 3.5 FURTHER EXPERIMENTS

A natural question, given the improvements from pseudosaccades, is whether an 'ensemble of ensembles' would improve performance further? We started by looking further into the diversity of the saccade classifiers. Following Kuncheva & Whitaker (2003), we calculate the correlation between the saccade classifier errors and the baseline classifier errors using

$$\rho_{i,j} = \frac{N_{11}N_{00} - N_{01}N_{10}}{\sqrt{(N_{11} + N_{10})(N_{01} + N_{00})(N_{11} + N_{01})(N_{10} + N_{00})}}$$

, where $i$ is the base classifier, and $j$ is the saccade version with, and the definitions for $N_{00}, N_{01}, N_{11}$ and $N_{10}$ are given in table 5.

Table 3: Ensemble classifier accuracy for ensemble size $m = 5$ and projection dimensions $d \in \{450, 410, 350, 310, 250\}$, with the standard deviation from a sample of 50 ensembles. Values with '*' exceeded the top-$k$ accuracies of the baseline classifiers by at least 2 standard deviations

| Projection Dimensions | | Alexnet (%) | | GoogLeNet (%) | | ResNet-50 (%) | | ResNet-152 (%) | |
|---|---|---|---|---|---|---|---|---|---|
| | | Mean | Std Dev | Mean | Std Dev | Mean | Std Dev | Mean | Std Dev |
| 450 | Top-1 | 55.255 | 0.044 * | 65.918 | 0.051 * | 70.726 | 0.048 * | 72.826 | 0.043 * |
| | Top-3 | 72.235 | 0.033 * | 82.526 | 0.040 * | 85.743 | 0.037 * | 87.289 | 0.050 * |
| | Top-5 | 77.996 | 0.038 * | 87.207 | 0.038 * | 89.830 | 0.049 * | 90.901 | 0.032 * |
| 410 | Top-1 | 55.416 | 0.044 * | 65.910 | 0.047 * | 70.629 | 0.075 * | 72.759 | 0.065 * |
| | Top-3 | 72.342 | 0.041 * | 82.490 | 0.040 * | 85.730 | 0.051 * | 87.264 | 0.043 * |
| | Top-5 | 78.092 | 0.038 * | 87.194 | 0.036 * | 89.771 | 0.057 | 90.860 | 0.038 * |
| 350 | Top-1 | 55.520 | 0.051 * | 65.512 | 0.060 | 69.901 | 0.074 | 72.308 | 0.047 |
| | Top-3 | 72.294 | 0.056 * | 82.153 | 0.050 | 85.250 | 0.069 | 86.955 | 0.043 |
| | Top-5 | 78.061 | 0.045 * | 86.918 | 0.055 | 89.307 | 0.067 | 90.613 | 0.044 |
| 310 | Top-1 | 55.038 | 0.074 * | 64.596 | 0.068 | 68.793 | 0.103 | 71.269 | 0.064 |
| | Top-3 | 71.989 | 0.052 * | 81.429 | 0.047 | 84.286 | 0.080 | 86.253 | 0.047 |
| | Top-5 | 77.738 | 0.057 * | 86.261 | 0.051 | 88.512 | 0.069 | 90.072 | 0.037 |
| 250 | Top-1 | 53.299 | 0.076 | 61.188 | 0.071 | 65.511 | 0.158 | 68.588 | 0.171 |
| | Top-3 | 70.487 | 0.046 | 78.555 | 0.063 | 81.596 | 0.102 | 84.113 | 0.112 |
| | Top-5 | 76.398 | 0.066 | 83.878 | 0.057 | 86.247 | 0.068 | 88.283 | 0.076 |

Table 4: The mean ensemble classifier accuracy for ensemble size $m = 10$ for projection dimensions $d \in \{450, 410, 350, 310, 250\}$, with the standard deviation from a sample of 50 ensembles. Values with '*' exceeded the top-$k$ accuracies of the baseline classifiers by at least 2 standard deviations

| Projection Dimensions | | Alexnet (%) | | Googlenet (%) | | Resnet-50 (%) | | Resnet-152 (%) | |
|---|---|---|---|---|---|---|---|---|---|
| | | Mean | Std Dev | Mean | Std Dev | Mean | Std Dev | Mean | Std Dev |
| 450 | Top-1 | 55.348 | 0.035 * | 65.987 | 0.038 * | 70.828 | 0.032 * | 72.894 | 0.026 * |
| | Top-3 | 72.307 | 0.032 * | 82.582 | 0.036 * | 85.802 | 0.028 * | 87.365 | 0.029 * |
| | Top-5 | 78.081 | 0.030 * | 87.266 | 0.031 * | 89.903 | 0.035 * | 90.953 | 0.024 * |
| 410 | Top-1 | 55.568 | 0.048 * | 66.003 | 0.041 * | 70.734 | 0.055 * | 72.863 | 0.043 * |
| | Top-3 | 72.435 | 0.031 * | 82.568 | 0.034 * | 85.828 | 0.026 * | 87.361 | 0.028 * |
| | Top-5 | 78.195 | 0.035 * | 87.278 | 0.033 * | 89.871 | 0.039 * | 90.938 | 0.025 * |
| 350 | Top-1 | 55.716 | 0.048 * | 65.672 | 0.041 * | 70.065 | 0.047 | 72.500 | 0.040 |
| | Top-3 | 72.444 | 0.035 * | 82.310 | 0.036 | 85.385 | 0.048 | 87.118 | 0.025 |
| | Top-5 | 78.233 | 0.036 * | 87.069 | 0.033 * | 89.460 | 0.046 | 90.747 | 0.025 * |
| 310 | Top-1 | 55.300 | 0.068 * | 64.866 | 0.043 | 69.038 | 0.054 | 71.497 | 0.043 |
| | Top-3 | 72.203 | 0.055 * | 81.660 | 0.045 | 84.524 | 0.073 | 86.457 | 0.036 |
| | Top-5 | 77.966 | 0.045 * | 86.494 | 0.044 | 88.733 | 0.054 | 90.304 | 0.025 |
| 250 | Top-1 | 53.643 | 0.050 | 61.604 | 0.055 | 65.864 | 0.121 | 68.956 | 0.093 |
| | Top-3 | 70.803 | 0.046 | 78.955 | 0.056 | 81.955 | 0.073 | 84.484 | 0.082 |
| | Top-5 | 76.728 | 0.046 | 84.268 | 0.048 | 86.639 | 0.071 | 88.624 | 0.066 |

In table 6, we see that – based on this summary statistic – the accuracy of the saccade classifiers is highly correlated with that of the corresponding baseline classifier, indicating to us that the classifier performance is not greatly reduced by pseudosaccade projection. Table 7 meanwhile shows that although the saccade classifier errors are correlated with one another, this is to a lesser degree than to the baseline classifiers. These facts suggest that there might be little to gain from combining the

Table 5: 2x2 table of the relationship between the classifiers $D_i$ and $D_j$

| | $D_j$ Correct | $D_j$ Wrong |
|---|---|---|
| $D_i$ Correct | $N_{11}$ | $N_{10}$ |
| $D_i$ Wrong | $N_{01}$ | $N_{00}$ |

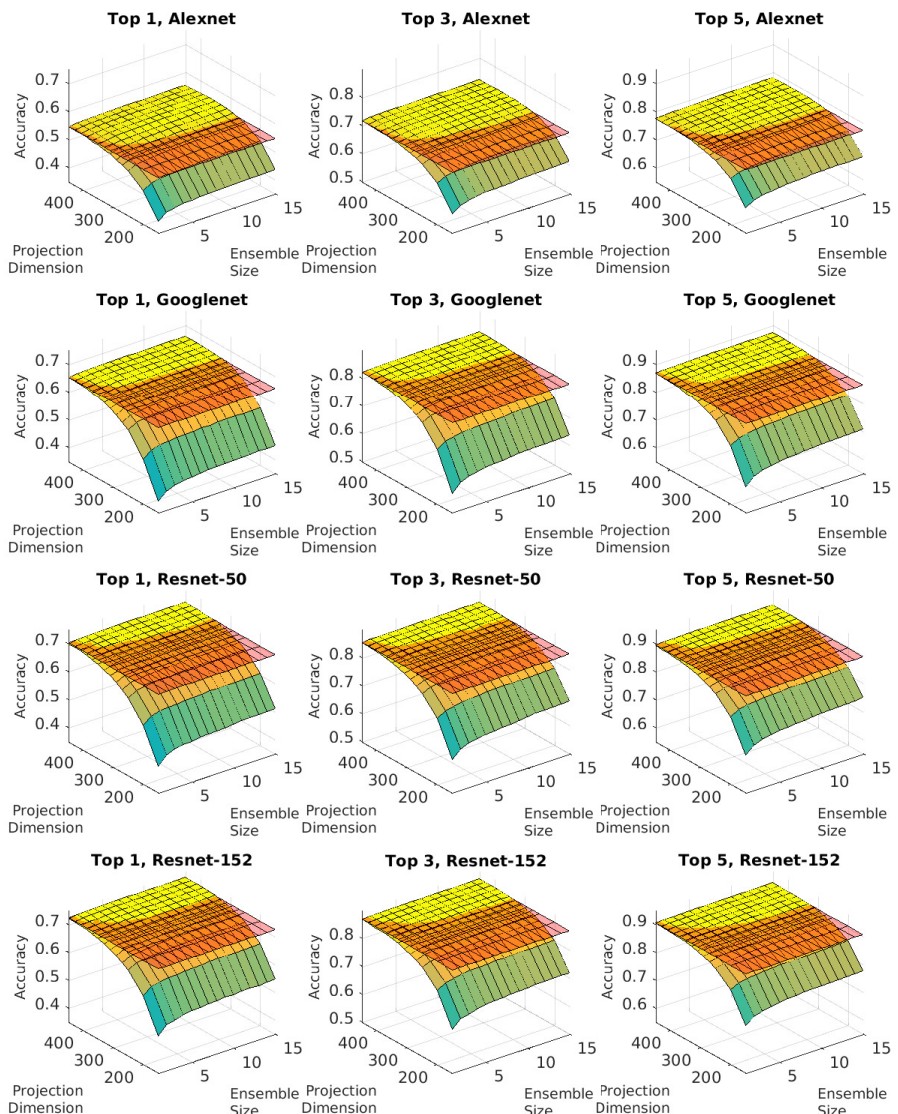

Figure 2: Accuracy vs ensemble size and projection dimension. Reference plane shows the accuracy for the baseline classifier

pseudosaccade ensembles from different DNNs into a larger ensemble. However since all of the accuracies are already high it seemed worthwhile to examine where the improvements were coming from - were these for similar class labels for every classifier for example?

Digging deeper we observed that the classification accuracy of the individual classes is not uniformly affected by pseudosaccades. Moreover, at this lower level of granularity we see that the different architectures do tend to be affected by the pseudosaccades differently.

Tables 10,11 and 9 show lists of predicted class labels for a given class label for ResNet-152, with projection dimension 390 and ensemble size 5. Note that there are 50 instances in each of the

true class labels, and we omitted predicted labels where there was only a single prediction or two predictions for reasons of space and readability.

In table 10, we present a list of labels for which the ResNet-152 classifier obtained less than 20% recall – considering this and similar tables for AlexNet and GoogLeNet (omitted due to space constraints) we observed that the ensemble of pseudosaccade classifiers performs similarly to the baseline classifier on labels that are also difficult for the baseline classifier to predict accurately, but we also saw that the different classifier architecture have their own sets of 'difficult labels' that are different.

Finally tables 9 and 11 give examples of class labels where the ensemble classifier respectively gives either a large improvement or is much worse ($\pm 10\%$) on classification accuracy for these classes. Comparing with the other architectures we found that the saccade classifiers were affected differently on different classifier architecture.

Thus, although high-level summary statistics seemed to indicate little diversity between the different ensemble classifiers, a more principled investigation reveals that the errors for both the original DNNs and for the corresponding pseudosaccade ensembles arise from different classes and different instances in the dataset.

We therefore constructed two ensemble classifiers - one using the four baseline DNNs and one which combined four pseudosaccade ensembles to see if further improvements were possible. We used five-fold cross-validation on the validation set data to train a shallow neural network with a single hidden layer with ReLU activations on the baseline scores for 40000 images from the validation dataset to learn a weighting function for the ensemble of baseline classifiers. We used the average and maximum scores from pseudosaccade versions of the four DNNs for the same 40000 images to train a similar network to weight the 'ensemble of ensembles'. We evaluated both ensembles using the 10000 remaining held-out images from the validation dataset and estimated the top-1,top-3 and top-5 accuracies for both ensembles with the cross-validation error. We carried out one round of five-fold cross-validation for the baseline classifiers and 50 rounds for the pseudosaccade classifiers, for different $d, m, k$ triples and calculated the mean accuracies and their standard deviations. For both types of ensemble we saw substantial improvements over the original baseline accuracies and, consistent with our earlier experiments, the pseudosaccade ensembles were yet again able to outperform the ensemble of baseline DNN classifiers. Figure 3, shows the accuracy of the DNN ensemble versus the pseudosaccade ensembles for different $d, m, k$ triples. The horizontal orange plane indicates the (average) accuracy of the DNN ensemble. The pseudosaccade ensembles outperforms the increased Top-1 accuracy baseline of 75.78% by 0.3%, and the accuracy of the best performing classifier Resnet-152 by 3.7%. We conjecture that further, possibly minor, improvements in accuracy may be possible using a more careful approach to learn the weighting function.

Table 6: The average classifier correlation $\rho_{base,saccade}$ between baseline classifier and saccade classifiers

| Saccade Dimensions | AlexNet | GoogLeNet | Resnet-50 | Resnet-152 |
|---|---|---|---|---|
| 450 | 0.8959 | 0.8952 | 0.8916 | 0.8949 |
| 430 | 0.8851 | 0.882 | 0.877 | 0.8811 |
| 410 | 0.8753 | 0.8694 | 0.8615 | 0.8681 |
| 390 | 0.865 | 0.8552 | 0.8479 | 0.8552 |
| 370 | 0.8524 | 0.8368 | 0.8301 | 0.8377 |
| 350 | 0.8375 | 0.8164 | 0.8078 | 0.8164 |
| 330 | 0.8221 | 0.7937 | 0.7853 | 0.7943 |
| 310 | 0.8039 | 0.7683 | 0.7601 | 0.7715 |
| 290 | 0.785 | 0.7418 | 0.7335 | 0.7449 |
| 270 | 0.764 | 0.7119 | 0.7068 | 0.7174 |
| 250 | 0.742 | 0.678 | 0.676 | 0.6889 |
| 200 | 0.6727 | 0.5729 | 0.5806 | 0.5977 |
| 150 | 0.5626 | 0.437 | 0.4563 | 0.4666 |

Table 7: The average classifier correlation $\rho_{saccade_1,saccade_2}$ between all pairs of saccade classifiers

| Saccade Dimensions | AlexNet | GoogLeNet | ResNet-50 | ResNet-152 |
|---|---|---|---|---|
| 450 | 0.8847 | 0.8830 | 0.8809 | 0.8876 |
| 430 | 0.8744 | 0.8702 | 0.8682 | 0.8754 |
| 410 | 0.8657 | 0.8597 | 0.8562 | 0.8648 |
| 390 | 0.8564 | 0.8484 | 0.8456 | 0.8543 |
| 370 | 0.8448 | 0.8344 | 0.8337 | 0.8419 |
| 350 | 0.8320 | 0.8202 | 0.8196 | 0.8266 |
| 330 | 0.8196 | 0.8060 | 0.8059 | 0.8121 |
| 310 | 0.8077 | 0.7917 | 0.7938 | 0.7991 |
| 290 | 0.7956 | 0.7771 | 0.7792 | 0.7849 |
| 270 | 0.7835 | 0.7621 | 0.7667 | 0.7700 |
| 250 | 0.7717 | 0.7466 | 0.7528 | 0.7572 |
| 200 | 0.7369 | 0.7027 | 0.7138 | 0.7211 |
| 150 | 0.6942 | 0.6473 | 0.6666 | 0.6750 |

Table 8: Top-$k$ accuracies for the "ensemble of ensembles" for ensemble size $m \in \{1, 5, 10, 15\}$, projection dimensions $d \in \{450, 410, 350, 310, 250\}$. Entries with '*' denotes statistically significant improvements over the baseline accuracies (top-1: 75.78%, top-3: 87.65%, top-5: 90.35%)

| Projection Dimensions | | One Saccade Mean | Std Dev | 5 Saccades Mean | Std Dev | 10 Saccades Mean | Std Dev | 15 Saccades Mean | Std Dev |
|---|---|---|---|---|---|---|---|---|---|
| 450 | Top-1 | 75.73 | 0.12 | 76.08 | 0.09 * | 76.16 | 0.05 * | 76.17 | 0.05 * |
| | Top-3 | 87.51 | 0.10 | 87.98 | 0.07 * | 88.13 | 0.05 * | 88.21 | 0.04 * |
| | Top-5 | 90.19 | 0.08 | 90.74 | 0.07 * | 90.86 | 0.05 * | 90.92 | 0.04 * |
| 410 | Top-1 | 75.55 | 0.14 | 75.96 | 0.08 * | 76.04 | 0.06 * | 76.10 | 0.05 * |
| | Top-3 | 87.38 | 0.11 | 87.99 | 0.07 * | 88.15 | 0.05 * | 88.22 | 0.03 * |
| | Top-5 | 90.12 | 0.08 | 90.76 | 0.07 * | 90.91 | 0.06 * | 90.97 | 0.05 * |
| 350 | Top-1 | 74.65 | 0.15 | 75.36 | 0.11 | 75.51 | 0.08 | 75.57 | 0.06 |
| | Top-3 | 86.83 | 0.09 | 87.75 | 0.07 | 87.95 | 0.06 * | 88.06 | 0.05 * |
| | Top-5 | 89.61 | 0.12 | 90.57 | 0.08 * | 90.79 | 0.07 * | 90.91 | 0.05 * |
| 310 | Top-1 | 73.58 | 0.15 | 74.56 | 0.11 | 74.73 | 0.08 | 74.80 | 0.06 |
| | Top-3 | 85.91 | 0.12 | 87.06 | 0.09 | 87.31 | 0.07 | 87.46 | 0.05 |
| | Top-5 | 88.92 | 0.13 | 90.13 | 0.10 | 90.50 | 0.07 * | 90.67 | 0.06 * |
| 250 | Top-1 | 70.34 | 0.19 | 71.78 | 0.11 | 71.96 | 0.11 | 72.01 | 0.09 |
| | Top-3 | 83.33 | 0.15 | 85.04 | 0.09 | 85.35 | 0.08 | 85.48 | 0.06 |
| | Top-5 | 86.65 | 0.17 | 88.44 | 0.11 | 88.89 | 0.09 | 89.08 | 0.06 |

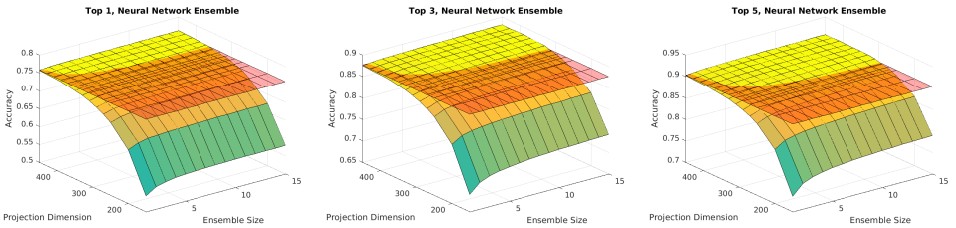

Figure 3: Top-1, top-3 and top-5 accuracy of the "ensemble of ensembles" vs ensemble size and projection dimension

Table 9: Labels where ensemble method performed significantly better ($\geq 10\%$) than the baseline Resnet-152 Imagenet classifier. Number of instances for which the given label was returned by classifier in brackets.

| true label | base label | saccade label | ensemble label |
|---|---|---|---|
| **rock crab** | dungeness crab (4)
**rock crab (28)**
hermit crab (4) | **rock crab (30)**
crayfish (3)
hermit crab (3) | dungeness crab (3)
**rock crab (33)**
hermit crab (4) |
| **bedlington terrier** | **bedlington terrier (28)** | **bedlington terrier (42)** | **bedlington terrier (43)** |
| **labrador retriever** | bloodhound (3)
saluki (3)
golden retriever (3)
**labrador retriever (34)** | saluki (3)
**labrador retriever (36)** | **labrador retriever (39)** |
| **bell cote** | **bell cote (26)**
chime (3)
church (11)
monastry (4) | **bell cote (31)**
church (8)
monastry (4) | **bell cote (31)**
church (9)
monastry (4) |
| **bow** | **bow (30)** | **bow (32)** | **bow (35)** |
| **necklace** | chain (3)
**necklace (40)** | **necklace (46)** | **necklace (46)** |
| **pitcher** | **pitcher (19)**
vase (4)
water jug (6) | **pitcher (25)**
vase(4)
water jug (5) | **pitcher (24)**
vase (3)
water jug (8) |
| **plastic bag** | **plastic bag (24)** | **plastic bag (26)** | **plastic bag (29)** |
| **hen of the wood** | coral fungus (3)
**hen of the wood (35)** | coral fungus (4)
**hen of the wood (36)** | **hen of the wood (40)** |

Table 10: Labels where Resnet-152 Imagenet classifier achieved $\leq 20\%$ recall. Number of instances for which the given label was returned by classifier in brackets.

| true label | base label | saccade label | ensemble label |
|---|---|---|---|
| **cassette player** | **cassette player (10)**
cd player (4)
radio (3)
tape player (22) | **cassette player (9)**
cd player (4)
tape player (23) | **cassette player (9)**
cd player (4)
radio (3)
tape player (21) |
| **crt screen** | desk (6)
desktop computer (8)
monitor (4)
**crt screen (8)**
television (8) | desk (6)
desktop computer (8)
monitor (5)
**crt screen (9)**
television (9) | desk (5)
desktop computer (8)
laptop computer (3)
monitor (6)
**crt screen (9)**
television (9) |
| **sunglass** | **sunglass (11)**
sunglasses (19) | **sunglass (10)**
sunglasses (19) | **sunglass (11)**
sunglasses (16) |

## 4 CONCLUSIONS AND FUTURE WORK

We demonstrated that using a very simple, and computationally cheap, 'pseudosaccade' ensemble learning approach can improve the image classification performance of DNNs. This improvement is small but statistically significant at the 5% level and requires no complicated training or optimization, simply selecting two integer parameters $d$ and $m$. In our experiments setting $d \geq 350$ and $m \geq 5$ worked well, with the approach much more sensitive to over-small values of $d$ than of $m$. Following a careful analysis of the sources of error in our classification problem, we showed that these improvements also propagate to a weighted ensemble of pseudosaccade versions of (off-the-shelf) DNNs. An open problem is whether a (simple, or low overhead) non-uniform sampling scheme for constructing pseudosaccade data exists that could improve performance further, possibly mediated by a scene-dependent prior. Human visual processing suggests that such a scheme should be at least a possibility. We are examining non-uniform sampling schemes such as stratified sampling, and also techniques such as seam-carving, with a view to progress in this direction.

Table 11: Labels where ensemble method performed significantly worse ($\geq 10\%$) than the baseline Resnet-152 Imagenet classifier. Number of instances for which the given label was returned by classifier in brackets.

| true label | base label | saccade label | ensemble label |
|---|---|---|---|
| **mantis** | walking stick (3)
**mantis (37)** | walking stick (5)
**mantis (31)** | walking stick (5)
**mantis (32)** |
| **abaya** | **abaya (41)** | **abaya (37)**
cloak (3) | **abaya (36)**
cloak (3) |
| **perfume** | **perfume (40)** | **perfume (35)** | **perfume (34)** |
| **wok** | **wok (28)**
hot pot (10) | dutch oven (4)
frying pan (3)
**wok (22)**
hot pot (9) | frying pan (4)
**wok (23)**
hot pot (11) |

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
