# OpenReview forum: "Pseudosaccades: A simple ensemble scheme for improving classification performance of deep nets"
_ICLR.cc/2019/Conference_

### Official Review · AnonReviewer1 · 2018-11-02
**Pseudosaccades**

**Rating:** 4
**Confidence:** 5

**Review:**

The paper proposes a data augmentation technique where the input image is sub-sampled by randomly sampling rows and columns without replacement, which the authors call ‘pseudosaccades’. Rather than multiple classifiers, the authors ensemble using multiple ‘pseudosaccades’ as input, with the same network.

Comments:
I think that the proposed augmentation is a neat trick. However, the inner-workings of the method are poorly presented (or not well understood). For eg. In section 3.5, while discussing the effects of the method on individual classes, the authors mention ‘different architectures do tend to be affected by the pseudosaccades differently’ and provide no further insights.

There are no experiments that compare this method with other standard data augmentation techniques. For instance, one could use a similar ensembling technique for transformations like shear, translation, rotation, etc. by randomly sampling their corresponding parameters. I would be interested in experimental results that compare the proposed ensemble with ensembles constructed using these common techniques.

Since there is no reason for this technique to be used in isolation (I found no such motivation in the paper), it would be insightful to have experimental results where this technique is combined with the aforementioned standard augmentation techniques. Will this method’s impact on the accuracy change with these other augmentations? (Ablation studies would be useful).

This is a a form of regularization and can be thought of reverse structured dropout. Also have the authors compared this with Cutout [1, 2]? Similar experiments and comparisons would be insightful.

[1] Terrance DeVries and Graham W Taylor. Improved regularization of convolutional neural networks with cutout. arXiv preprint arXiv:1708.04552, 2017.
[2] Zhun Zhong, Liang Zheng, Guoliang Kang, Shaozi Li, and Yi Yang. Random erasing data augmentation. arXiv preprint arXiv:1708.04896, 2017.

In summary:
The performance improvements are incremental. The paper lacks sufficient technical contribution. Further, it does not provide comparisons with standard techniques and similar augmentation methods to demonstrate the usefulness of the method.

---

### Official Review · AnonReviewer3 · 2018-11-03
**numbers, comparisons, and pratical value**

**Rating:** 4
**Confidence:** 4

**Review:**

	This paper proposes a data ensemble method for image classification: sub sample an image, classify each sub sample, and vote those sub samples to get the final decision.

	Questions:
	1. The validation accuracy of ResNet is much lower than that reported in the original ResNet paper, https://arxiv.org/pdf/1512.03385.pdf. For example, the top-1/5 accuracy of RestNet-50 is 79+/94+, which is only about 70+/89+% in this paper. Similarly, there is a big gap between the results of ResNet-152 reported in this paper and the original ResNet paper. Maybe I misunderstand something; otherwise, the results in this paper are not reliable.
	2. This work does not compare with any other data augmentation methods for testing, e.g., the widely used 10-crop test [ref1, ref2]: “At test time, the network makes a prediction by extracting five 224 × 224 patches (the four corner patches and the center patch) as well as their horizontal reflections (hence ten patches in all), and averaging the predictions made by the network’s softmax layer on the ten patches.”
        3. A minor issue is the practical value of this work. If one can afford the computational cost of data ensemble test, why not train m (as the data ensemble in this paper) models and ensemble them given that ensemble of models usually brings more accuracy improvement? Note that the training of multiple models is conducted offline and the ensemble of models is of the same computation cost compared with the method in this paper.

[ref1] Krizhevsky, Alex, Ilya Sutskever, and Geoffrey E. Hinton. "Imagenet classification with deep convolutional neural networks." Advances in neural information processing systems. 2012.
[ref2] He, Kaiming, et al. "Deep residual learning for image recognition." Proceedings of the IEEE conference on computer vision and pattern recognition. 2016.

---

### Official Review · AnonReviewer2 · 2018-11-05
**low technical novelty, but interesting results**

**Rating:** 5
**Confidence:** 4

**Review:**

Pros:
-- Superior empirical results are the key highlights of this paper.
-- The experiments are well designed and benchmarked against the state-of-the-art models.

Cons:
-- One typically uses affine transformations of the training images to improve the performance of the CNN. From that perspective, the paper does not offer any new insight. I am not entirely convinced that this is a novel enough contribution to be accepted in ICLR.
-- The "ensemble of ensembles" approach described in Section 3.5 is not clear.
-- Overall, the paper does not have much novelty, but the results are quite promising.

---

### Meta-Review · Area_Chair1 · 2018-12-14

**Confidence:** 4
**Recommendation:** Reject

**Metareview:**

The paper proposes a data augmentation technique to ensemble classifiers.
Reviewers pointed to a few concerns, including a lack of novelty, a lack
of proper comparison with state-of-the-art models or other data augmentation
approaches.
Overall, all reviewers recommended to reject the paper, and I concur with them.